# Mixed Heterotopic Gastrointestinal/Respiratory Oral Cysts in Newborns: From Prenatal Diagnosis to Histopathological and Therapeutic Management: A Case Report and Literature Review

**DOI:** 10.3390/diagnostics14030339

**Published:** 2024-02-04

**Authors:** Valentin Nicolae Varlas, Ioanina Parlatescu, Dragos Epistatu, Oana Neagu, Roxana Georgiana Varlas, Laura Bălănescu

**Affiliations:** 1Department of Obstetrics and Gynecology, Filantropia Clinical Hospital, 011132 Bucharest, Romania; valentin.varlas@umfcd.ro (V.N.V.); roxana-georgiana.bors@drd.umfcd.ro (R.G.V.); 2Faculty of Dentistry, Carol Davila University of Medicine and Pharmacy, 010221 Bucharest, Romania; 3Department of Oral Pathology, Faculty of Dentistry, Carol Davila University of Medicine and Pharmacy, 17-21, Calea Plevnei Street, 020021 Bucharest, Romania; 4Department of Radiology, Faculty of Dentistry, Carol Davila University of Medicine and Pharmacy, 17-21, Calea Plevnei Street, 020021 Bucharest, Romania; 5Department of Anatomopathology, Carol Davila University of Medicine and Pharmacy, 050474 Bucharest, Romania; oana.neagu@yahoo.com; 6Department of Pediatric Surgery, Children Emergency Hospital “Grigore Alexandrescu”, 011743 Bucharest, Romania; laura.balanescu@umfcd.ro; 7Faculty of General Medicine, Carol Davila University of Medicine and Pharmacy, 050474 Bucharest, Romania

**Keywords:** lingual cyst, heterotopic oral gastrointestinal cyst, foregut duplication cyst, lingual choristoma, lingual cyst with a respiratory component, prenatal diagnosis, differential diagnosis, treatment, prognosis

## Abstract

Fetal lingual tumors are very rare, and their early prenatal diagnosis is important for defining the subsequent therapeutic strategy. In this study, we aimed to describe a case of a congenital septate lingual cyst and perform an extensive literature review on two main databases (PubMed, Web of Science), analyzing the clinical manifestations, the imaging appearance, the differential diagnosis, and particularities regarding the treatment of these tumors. The electronic search revealed 17 articles with 18 cases of mixed heterotopic gastrointestinal/respiratory oral epithelial cysts that met the eligibility criteria and were included in this review. The clinical case was diagnosed prenatally during second-trimester screening. On the eighth day of life, the fetus underwent an MRI of the head, which revealed an expansive cystic process on the ventral side of the tongue with the greatest diameter of 21.7 mm, containing a septum of 1 mm inside. On the 13th day of life, surgery was performed under general anesthesia, and the lingual cystic formation was completely excised. The postoperative evolution was favorable. The histopathological examination revealed a heterotopic gastric/respiratory-mixed epithelial cyst with non-keratinized respiratory, gastric squamous, and foveolar epithelium. The lingual cyst diagnosed prenatally is an accidental discovery, the differential diagnosis of which can include several pathologies with different degrees of severity but with a generally good prognosis.

## 1. Introduction

Like other congenital oral cavity tumors, lingual cystic formations are discovered incidentally in the prenatal period, especially during the second-trimester ultrasound [1]. A possible mechanism of occurrence is given by the sequestration of respiratory or gastric differentiated pluripotent embryonic remains in the tongue’s structure during its development, with subsequent cystic degeneration. In contrast to classic foregut duplication cysts that contain intramural smooth muscle and mucosa, heterotopic cysts may associate other epithelial and mucosal histological components such as gastric, respiratory, pancreatic-type, squamous, columnar, and cuboidal components or combinations of them.

These congenital anomalies are rare; lingual or floor-of-the-mouth localization represents only 0.18% of all gastrointestinal duplication cysts, according to the study by Gantwerker et al. [2]. The prenatal identification of these cysts is made predominantly using an ultrasound. An MRI is important to evaluate the characteristics of these masses and their relationship with the pharyngeal structures and airways. The real-time prenatal ultrasound visualization of the oral cavity also allows the evaluation of swallowing movements and, respectively, the sucking reflex.

Most cysts may be asymptomatic or have different clinical manifestations, depending on their topography and size. Intrapartum therapeutic management is conditioned by the onset of the respiratory distress syndrome of the newborn or, subsequently, by the impossibility or difficulty of breastfeeding. The presence of masses in the oral cavity is very difficult to appreciate due to its possible technical limitations during a routine ultrasound [3].

Postnatally, endotracheal intubation is sometimes necessary until a cyst reduction procedure or surgical resection with its anatomopathological evaluation is performed [4]. Due to the existence of some sporadic cases in the literature, there is no consensus regarding the optimal time for their therapeutic solution, the frequency of monitoring, the interpretation of ultrasound findings, the intrapartum strategy, the possible postnatal complications, such as those during breastfeeding, as well as the histological type [5,6].

In this article, we report a case of a mixed heterotopic oral gastrointestinal and respiratory cyst diagnosed prenatally, as well as a literature review regarding the diagnosis, therapeutic solution, and prognosis of these cases. Counseling in these cases and the involvement of a multidisciplinary team can represent the key to successful treatment.

## 2. Case Report

We present the case of a term male newborn to a 30-year-old mother who was followed throughout the pregnancy in our hospital. She had one previous pregnancy with a child with a complex cardiac malformation (tricuspid atrophy, single ventricle).

At 23 weeks of pregnancy, during the second-trimester screening, the presence of an orofacial cystic lesion with anterior lingual localization was discovered. The lesion was an anechoic, homogenous, septate cystic formation without solid areas, with well-defined edges and thin walls, measuring 2 × 0.8 × 0.5 cm. No Doppler signal was evident at this level. The cyst was mobile with tongue movements, and the tongue was slightly open, with the presence of swallowing movements (Figure 1a–c). No other ultrasound abnormalities or polyhydramnios were observed. The non-invasive fetal DNA test was performed and showed a normal karyotype.

In subsequent ultrasounds, no significant growth of this formation was observed, and the amniotic fluid was normal, which did not require the performance of an intrauterine intervention. The last ultrasound performed at 36 weeks highlighted the presence of swallowing movements (Appendix A) and the lack of an expansive process at the oropharyngeal level (Figure 1d).

The pregnant woman gave birth at 38 weeks + 4 days by cesarean section to a male child weighing 3450 g. Apgar scores were 7 at birth and after 1 min and 9 after 5 min. Later, the newborn was admitted to the neonatal intensive care unit (NICU), where he did not require endotracheal intubation.

After birth, a neonatal examination with direct laryngoscopy revealed an enlarged tongue due to the cystic mass on the anterior face of the tongue and no palatal or epiglottic changes.

Postnatally, on the eighth day of life, a preoperative MRI evaluation of the newborn was performed, which confirmed the absence of an extension at the level of the oropharynx (Figure 1e,f).

The clinical examination at birth revealed only lingual malformation, highlighting the extraoral component in the anterior part of the tongue, which comprised a septate cystic component (Figure 2a,b).

On the 13th day of life, the child underwent plastic surgery for complete cyst excision, strictly limited to the tongue without extension at its base. Intraoperatively, the cystic content was represented by mucoid fluid. A macroscopic description revealed a tissue fragment measuring 2.7 × 1.7 × 1.5 cm with cystic formation measuring 2 × 1 cm, empty of content. A microscopic description showed fragments of lingual tissue with intramuscular cystic formation, bilocular, lined with a ciliated respiratory-type columnar epithelium, non-keratinizing squamous epithelium, and mucinous columnar epithelium similar to the gastric-foveolar epithelium. Cyto-nuclear atypia was not present, and the excision margins were free. Histological diagnosis found a benign cystic formation of the congenital lingual cyst type with respiratory, a non-keratinizing squamous, and gastric-foveolar epithelium (Figure 3a–c). Thus, the histological aspect advocates the diagnosis of heterotopic gastric/respiratory-mixed epithelial cysts.

The follow-up at 8 months showed a normal tongue from an aesthetic and functional point of view and no signs of recurrence (Figure 2c). The family and the pediatrician mentioned that he closes his mouth sometimes but keeps it open more often.

## 3. Discussion

To better evaluate the lingual cystic formations of the fetus and, respectively, the newborn, we performed a comprehensive search of the literature across all time, accessing the following two electronic databases: PubMed and Web of Science. The selected articles were written in English, using the following methodological approach: systematic analyses, case series, and case reports. The search focused on clinical, imaging, and histological diagnostic, therapeutic, and prognostic criteria, using the following search strategy with MeSH terms: “tumors/oral cystic lesions”, “lingual cyst”, “foregut duplication cyst”, “lingual choristoma”, “heterotopic oral gastrointestinal cyst”, “lingual cyst with a respiratory component”, “prenatal diagnosis”, “treatment”, “prognosis”, “differential diagnosis” AND “fetuses” OR “neonates”. A total of 17 articles with 18 cases met the eligibility criteria and are included in the review of this article (Table 1).

Following the analysis of the 18 cases identified, after the removal of the 3 cases where no data were available regarding the sex of the newborn, the predominance of the female sex was observed, the female/male ratio being 1.5:1, contrary to data from the literature that attests a male predominance. In relation to cyst topography, 9 are described on the floor of the mouth and tongue, and 9 are anterior/on the tip of the tongue.

Ten out of eighteen (55.5%) patients were asymptomatic, 4 (22.2%) cases presented breastfeeding difficulties, and 4 (22.2%) had respiratory difficulties. The mean age at surgery was 32.1 days, and the mean largest diameter of the lingual cyst was 2.58 cm. The surgical procedure was complete excision in 17 (94.4%) cases and incomplete in 1 case, the prognosis being favorable in most cases. It should be noted that in one case, the surgical approach was direct excision with the osteotomy of the affected mandible, and in another case, partial glossectomy. Two cases required the intrapartum ex utero (EXIT) procedure technique with intubation/tracheostomy of the newborn with the maintenance of fetal circulation.

Prenatal diagnosis of congenital fetal oral tumors is essential for favorable perinatal outcomes. Changes in the stage of embryogenesis can determine these congenital anomalies, which can appear as cystic, solid, or mixed masses. Imaging represents the basic element in establishing the diagnosis, the therapeutic conduct, and the prognosis of these cases. Ultrasound is quick and easy to perform and is useful for the first-line investigation of cervical masses in newborns and infants. CT is rarely used, reserved for cases with cysts with mandibular extension, due to the advantage of displaying the bone structure. An MRI scan is superior to CT in determining the components and extent of the cystic mass and in establishing its relationship with the surrounding blood vessels and soft tissue. However, MRI is limited when the lesion is mobile due to movement or respiratory artifacts.

In conclusion, the ultrasound is essential in establishing the correct diagnosis, but the MRI examination has become mandatory, as data provided regarding the soft tissues are superior in treatment and evolution [15,23].

Yan et al. showed that at 13 weeks during first-trimester screening, the earliest diagnosis of an oral mass was made. Most of these tumor formations were detected in the second trimester and fewer in the third trimester; their occurrence was not correlated with a certain trimester of pregnancy [3].

The 3D ultrasound reconstruction of the fetal face can highlight anatomical changes regarding the extraoral part of the tumor formation that protrudes from the mouth. Doppler evaluation is usually important in the case of solid tumors by highlighting an abnormal vascular architecture [1,3]. The role of MRI is mainly to establish the compressive effect of the tumor mass on the hypopharynx, thus establishing the need for the orotracheal intubation of the newborn or esophageal obstruction [3,15]. A dynamic ultrasound evaluation is necessary to follow the growth of this formation and observe its relationships with the trachea, oropharynx, and nasopharynx during swallowing movements.

Carachi et al., in a retrospective study, carried out studies over 42 years on a group of 21 children with duplicate cysts of the tongue, identifying only one case that was diagnosed prenatally [24]. In another study, Kieran et al., on 12 patients with cystic lesions in the oral cavity, observed that only 1 patient was diagnosed prenatally with an ultrasound, the rest being detected postnatally mainly via an MRI and to a lesser extent using CT. In the same group, only one patient needed to be intubated [25].

The restrictive respiratory syndrome was not observed in cystic formations located on the tongue’s anterior side and measuring 1.5–2.4 cm in size [26]. Furthermore, prenatally, the existence of a large lingual cystic formation with a diameter >2.5 cm is found, and either an anesthetist or an otolaryngology specialist is needed to manage a possible airway obstruction. In our case, the maximum diameter of the lesion was 2.7 cm, and no endotracheal intubation was required.

In the case of lingual cystic malformations, prenatal genetic karyotyping is not mandatory because no associated anomalies have been identified. The accuracy of the histological evaluation is crucial due to the overlapping imaging and clinical characteristics of these cystic lesions on the tongue. For a better understanding and differentiation of the histological results obtained postoperatively to establish the correct diagnosis, the terminology approved by Manor et al. is still used [27].

A congenital oral cyst with respiratory epithelium is very rare and can be found on the tongue, floor of the mouth, and pharynx; 18 cases were reported in the literature in newborns less than 4 months after birth, of which 7 were detected using a prenatal ultrasound (4 cases in the second trimester and 3 cases in the third trimester) (Table 1). The histological component of the lining of the lingual cyst had either only the respiratory epithelial component or was represented by both types of epithelium (respiratory and gastric/intestinal) [28]. A possible etiological explanation of these cysts is given by the pluripotential undifferentiated cells, which, during their migration, can be caught in the pharyngeal arches involved in the development of the tongue, with subsequent differentiation in the respiratory epithelium [29]. Unlike histological analysis, immunohistochemistry studies can only demonstrate the shared appearance of respiratory epithelium with squamous epithelium [30]. Immunohistochemistry in the case of gastrointestinal and respiratory heterotopic cysts observed that epithelial cells are positive for cytokeratin (CK) 7 and thyroid transcription factor 1 and negative for CK20 [17]. In addition, in the case of the lymphoepithelial cyst, the lymphocytic inflammatory infiltrate was identified in 52.4% of cases and CD20+ cells in 57.2% [31].

Regarding terminology, heterotopic gastrointestinal cysts can have the following synonyms: a lingual duplication cyst, gastric heterotropia, choristoma, and enterocystoma. The presence of a respiratory and gastrointestinal epithelium in the cyst’s wall lining makes the differential diagnosis one of bronchogenic cysts [18].

The intra-lingual site of the thyroglossal duct cysts was found in 2.1% of patients, according to a systematic review including 47 cases. The authors also emphasized the need for thyroid function testing in addition to MRI data [32].

Dysontogenetic cysts represent a histological category as their walls are lined with keratinized, stratified squamous epithelium. According to the classification, they can be dermoid, epidermoid, and teratoid cysts [33].

A vallecular cyst is a rare and uncommon unilocular cystic lesion that carries the risk of upper airway obstruction, the primary clinical manifestation of which is the presence of stridor [34]. Vallecular cysts are part of major risk conditions, with an increased morbidity and mortality rate that require the rapid establishment of surgical intervention after diagnosis.

The differential diagnosis of cystic formations is carried out from a topographical and mainly histological point of view. Thus, a cyst located anteriorly, posteriorly, or on the floor of the mouth in the fetus may include the following diagnoses: ranula, a thyroglossal duct cyst, lymphoepithelial cyst, dysontogenetic cyst (dermoid cyst), heterotopic gastrointestinal and/or respiratory cyst, vallecular cyst, and hemangioma [5,6] (Figure 4).

The differential diagnosis of lingual cysts is very important in determining their treatment and monitoring (Table 2).

The evolution of these cystic masses can be spontaneous prenatal resorption [48] or a rapid increase in size that requires intrauterine intervention to prevent the obstruction of the upper airways, with a real fetal benefit. A series of studies evaluated the feasibility of cyst aspiration under ultrasound guidance either prenatally [49,50] or immediately after birth [21,51]. This procedure is debatable, the benefit being the case of large cysts that facilitate upper airway patency at birth. The main risk of well-vascularized tumors with a mixed solid/cystic component is bleeding in the upper airways when the needle is removed.

Tumor masses in the head and neck identified in newborns due to confusion about their origin require a clear diagnosis to help surgeons establish a therapeutic plan. Although the growth rate is low, these formations require imaging monitoring during pregnancy. The therapeutic management of these cases requires an appropriate preoperative evaluation, which includes a clinical examination, laboratory tests, and imaging scans (ultrasound, MRI, and CT) with or without sedation.

A prenatal ultrasound/MRI evaluation can identify the possible malposition of the fetal head, with the limitation of the flexion, an enlarged diameter of the trachea, the anterior displacement of the heart, a flattened diaphragm, or enlarged lungs with altered echogenicity. These factors can establish a possible degree of airway obstruction before birth, defining congenital high airway obstruction syndrome (CHAOS) [52]. An alternative in the therapeutic management of these cases is either the intrapartum ex utero (EXIT) technique [53,54] with the intubation/tracheostomy of the newborn with the maintenance of fetal circulation or the OOPS procedure with a placental vascular approach [55]. These techniques are performed during the cesarean section under general anesthesia, with partial or complete fetus extraction and uterine relaxation under tocolysis to ensure fetoplacental circulation. Thus, evaluating these cases requires a multidisciplinary team (obstetrician, neonatologist, pediatric surgeon, anesthetist, ± ear, nose, and throat—ENT specialist).

The surgical approach is based on the topography, size of the tumor, the obstruction of the upper airways, the difficulty of breastfeeding, and damage to the pharyngeal structures. An extraoral excision is preferred for large cysts in the submental area, and careful intraoral surgery in cases with extension to the infratemporal space or hypopharynx (sublingual cysts) [56,57]. Furthermore, the gastric epithelium inside these cysts can produce bleeding or ulceration. The immediate risks of delaying surgery in cases without airway/digestive obstructions are caused by trauma, hemorrhage, sepsis, and feeding difficulties, and late ones are generally functional (difficulty closing the mouth). The extraoral surgical approach with tongue reconstruction has the best functional results, with reduced risk of a vicious scar and tongue muscle damage, as demonstrated by our case’s favorable evolution [2,58]. Correcting the tongue’s position through exercises and patient counseling is important during the long-term follow-up.

Other techniques used, such as sclerotherapy, have not proven their effectiveness, and marsupialization has a risk of recurrence; thus, they have been abandoned [2]. However, in the case of vallecular cysts, marsupialization is as complete a surgical treatment as endoscopic excision, with a low recurrence rate [59]. The recurrence rate after surgical treatment is uncommon, and long-term follow-up is recommended.

The prognosis is generally favorable when potential cases with a restrictive appearance in the airways are identified prenatally, when no pharyngeal structures are affected, when no other congenital anomalies are associated, when surgical intervention is performed in the perinatal period, and with the complete excision of the cyst. Early diagnosis optimizes the therapeutic strategy of these cases to fulfill these conditions.

A particular aspect of this case was the presence of two cystic components diagnosed prenatally, which could have represented a surgical difficulty regarding the reconstruction of the tongue for aesthetic and functional results. Postoperatively, the evolution was favorable with the gradual resumption of oral and bottle feeding.

The limitations of this study are represented by the possible omission of some cases due to electronic access to two databases. Regarding possible directions for future research, there should be an aim to increase the early prenatal detection rate of these cases to develop optimal therapeutic management through the usual US/MRI imaging techniques associated with 3D reconstruction for a preoperative evaluation and the possible extension of the lingual cyst about the pharynx.

## 4. Conclusions

A lingual cyst is an unusual congenital anomaly that can be detected prenatally via ultrasound during second-trimester screening after 21 weeks of gestation and a complementary MRI at the end of the second trimester and in the third trimester to establish as accurately as possible the relationships of the cyst with the adjacent structures and especially with the pharynx. The prenatal evaluation of the oral cavity attests to the importance of identifying cystic masses due to the difficulty of establishing their differential diagnosis, which is mainly histopathological, as well as the possible complications that may occur at birth (the restriction of the respiratory tract and feeding difficulties) but with a generally good prognosis.

## Figures and Tables

**Figure 1 diagnostics-14-00339-f001:**
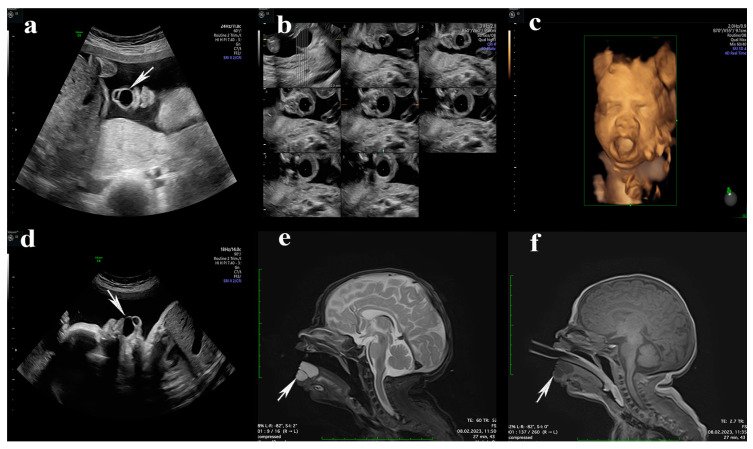
(**a**) The ultrasound image shows, at 23 weeks of gestation in the oral cavity, a bilocular cystic formation on the tongue (white arrow); (**b**) sections made using the TUI technique at the level of the tongue; (**c**) the 3D reconstruction of the lingual cyst formation; (**d**) sagittal incidence at 32 weeks to highlight the growth rate of the cyst (white arrow) and its relationship with the adjacent structures in the oropharynx; (**e**) postnatal, sagittal incidence of MRI showing an expansive cystic process, H-T2/H-T1, located on the ventral side of the tongue at approximately 21.7 × 18.2 × 15.8 mm with a septum of 1 mm and clear delimitation (white arrow), without signs of locoregional invasion, and without diffusion restriction. (**f**) The same incidence as endotracheal cannula exposure (white arrow show lingual cyst).

**Figure 2 diagnostics-14-00339-f002:**
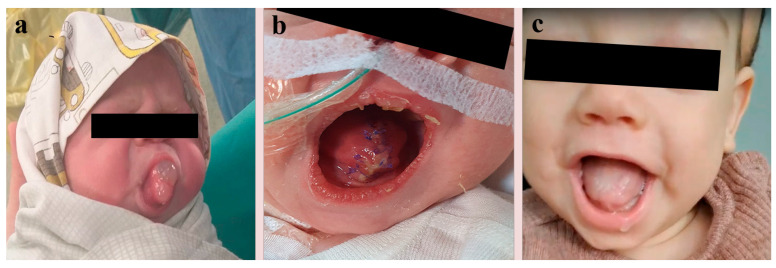
(**a**) The neonatal aspect of the tongue’s cyst at birth before surgical intervention, (**b**) the postoperative image of the tongue, (**c**) and aesthetic and functional appearance of the child’s tongue during a follow-up 8 months later.

**Figure 3 diagnostics-14-00339-f003:**
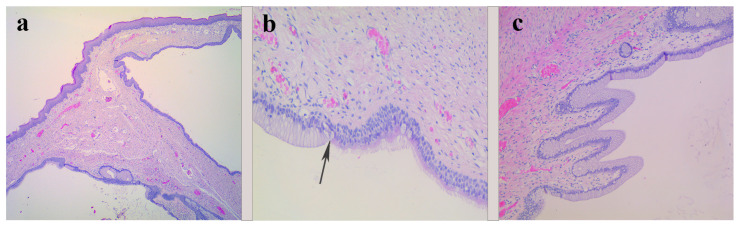
(**a**) Low power view depicting the lingual mucosa and a cystic lesion showing a septum and different types of epithelial lining, HE, 25×. (**b**) Cyst lining showing the transition between mucinous epithelium and respiratory epithelium (black arrow), HE, 200×. (**c**) The section of the lingual cyst with simple, short papillae covered by mucinous epithelium resembling gastric-foveolar-type epithelium, HE, 100×.

**Figure 4 diagnostics-14-00339-f004:**
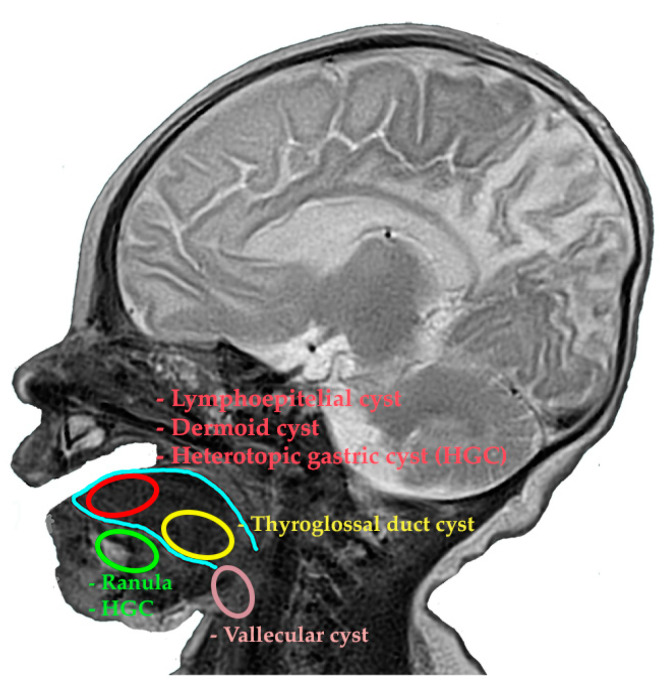
The main sites of lingual cystic formations using an MRI reconstruction.

**Table 1 diagnostics-14-00339-t001:** Overview of the heterotopic oral gastrointestinal and respiratory cysts in newborns for an all-time search in PubMed/WoS databases.

Author/Year	Age at Surgery	Sex	Lesion Site	Clinical Findings	Imagistic Findings	Cyst Size (cm)	Histopathology	Therapeutic Management	Outcome
Said-Al-Naief [7], 1999	2 m	M	The midline of the anterior floor of the mouth	Asymptomatic	MRI	2 × 1.6 × 1.8	Squamous and pseudostratified columnar epithelium	Complete excision	Favorable
Corić [8], 2000	2 m	M	Sublingual area	Breastfeeding difficulty	N/A	2.5 × 1.7 × 1.7	Squamous epithelium, intestinal type of epithelium and ciliated stratified columnar respiratory epithelium	Complete excision	N/A
Mandell [9], 2002	14 days	F	Anterior 2/3 of the tongue	Asymptomatic	CT	3.1 × 1.4 × 0.7	Non-keratinizing squamous, ciliated respiratory, and Gastric-foveolar epithelium	Incomplete excision	Favorable
Wetmore [10], 2002	6 days	F	Left floor of the mouth	Minor difficulty in feeding	MRI	2	gastric-type, respiratory epithelium, and skeletal muscle	Complete excision	Favorable
Noorchashm [11], 2004	Neonate	N/A	The anterior floor of the mouth with an intraosseous part (0.5 cm)	Asymptomatic	MRI/CT Single multiloculated mass extending from the chin to mandibular symphysis	N/A	Respiratory, gastric epithelium	Direct excision with osteotomy of the involved mandible	N/A
Hall [12], 2005	13 days	M	Base of tongue + floor of the mouth	Asymptomatic	Prenatal US (23 wks)/MRI Unilocular cyst	2.1 × 2.8 × 2.5	Squamous, respiratory, and gastric-type epithelium	Complete excision	Favorable
Leung [13], 2007	2 m	F	Floor of mouth	Difficulty in feeding	MRI	2.3 × 1.9 × 2.1	Glandular, respiratory epithelium	Complete excision	N/A
Hartnick [14], 2009	10 days	F	Sublingual space	Obstruction of larynx	Prenatal US (37 wks + 5 days) MRI Multicystic lesion	2.3 × 2.0 3.4 × 2.6 × 3.4	Squamous, respiratory, gastric-foveolar, and small-bowel epithelium	EXIT procedure Complete excision	First 6 m persistent tracheomalacia. Favorable after 10 m
Houshmand [15], 2011	6 days	F	Anterior 2/3 of the tongue	Respiratory difficulty—required nasal continuous positive airway pressure	Prenatal US (19 wks)/MRI (30 wks) Postnatal MRI (30 wks) Multiloculated cystic mass	3.2 × 2.6 × 2.8	Non-keratinizing squamous, ciliated respiratory, and gastric epithelium	Complete excision	Follow-up at 1 year—minor persistent oral defect + tongue protrusion
	2 m	F	Anterior 2/3 of the tongue	Asymptomatic	Prenatal US (20 wks) avascular multiloculated cyst/MRI (29 wks)	3.3 × 2.2 × 2.0	Squamous, respiratory, and gastric-foveolar epithelium	Complete excision	Favorable
Blanchard [16], 2012	Neonate	M	Tip of tongue	Asymptomatic	Prenatal US (32 wks)/MRI Lesion fluid, well limited, unilocular + polyhydramnios	2.4 × 1.7	Cylindrical, respiratory, malpighian epithelium	Complete excision	Favorable
Rosa [17], 2013	19 days	N/A	Anterior tongue	Asymptomatic	N/A	N/A	Ciliated columnar and non-keratinized squamous epithelium	Complete excision	6 m—no recurrence
Joshi [18], 2013	1 day	N/A	The ventral surface of the tongue	Asymptomatic	Postnatal MRI	1.7 × 0.7 × 1.0	Immature squamous, respiratory, gastric epithelium	Conservative management at 28 m—delayed surgery	Favorable
Gantwerker [2], 2014	6 days	F	Anterior tongue	Respiratory difficulty	Prenatal US (28 wks), MRI Unilocular, macrolobulated cyst	3.1 × 2.6 × 2.5	Squamous, gastric-foveolar, ciliated columnar, and cuboidal epithelium	Complete excision	Favorable
Luo [19], 2015	4 m	F	Floor of mouth	Asymptomatic	Postnatal MRI Unilocular cyst	2.5 × 2.1 × 1.8	Ciliated respiratory and gastric epithelium	Complete excision	Favorable
Knowles [20], 2017	2 m	M	Anterior 2/3 of the tongue	Asymptomatic	N/A	1.5 × 0.5	Respiratory, and gastric-foveolar epithelium	Complete excision	Favorable
Shabani [21], 2020	7 days	M	Floor of mouth and tongue	Respiratory difficulty	Prenatal US MRI Multiseptated mass + polyhydramnios	3 × 4 3.5 × 2.5 × 2.0	Squamous-lined cyst most consistent with epidermoid cysts	EXIT procedure Complete excision and partial glossectomy	8 m—mild protrusion of the tongue and open bite deformity
Arteta [22], 2021	5 days	F	Tip of tongue	Difficulty in mouth occlusion, suction, and breastfeeding	N/A	2	Squamous admixed with respiratory and antral-type epithelia	Complete excision	Favorable
Our case	13 days	M	Anterior 2/3 of the tongue	Asymptomatic	Prenatal US (23 wks) avascular Biloculated cyst/postnatal MRI	2.7 × 1.5 × 1.7	Respiratory, and gastric-foveolar epithelium	Complete excision	Favorable

M—male, F—female, US—ultrasound, MRI—magnetic resonance imaging, m—months.

**Table 2 diagnostics-14-00339-t002:** Clinical, imaging, histological, and treatment characteristics of lingual cysts.

	Ranula	Thyroglossal Duct Cyst	Lymphoepithelial Cyst	Dermoid Cysts/Dysontogenetic Cysts	Heterotopic Gastrointestinal and/or Respiratory Cyst	Vallecular Cyst
Incidence	0.74%	70% of all cervical masses	N/A	1:35,000 and 1:200,000	1:81,000	3.49–5.3:100,000
Predominance	Male	Male	Female 3:1	Male	Male 1.5–1.6:1	Male 3:1
Topography	Under the tongue/sublingual space/floor of the mouth	Cervical midline tongue base; posterior one-third of the tongue	Anterior two-thirds of the tongue; floor of the mouth; lateral border of the tongue; the ventral surface; soft and hard palates	Anterior two-thirds of the tongue; on the palatal or pharyngeal surface; sublingual mucosa	Anterior two-thirds of the tongue or the floor of the mouth (sublingual space)	The base of the tongue or lingual surface of epiglottis
Etiology	Obstruction of sublingual or minor salivary glands; imperforate salivary ducts in congenital ranula cyst	Partial or total persistence of the thyroglossal duct	Epithelial cells are included in lymphoid aggregates and desquamated epithelial lining obstructing the lymphatic tissue’s crypt aperture	Cyst filled with sebum-like material and an epidermal lining	Gastric tissues are entrapped in the oral cavity during fusion, resulting in midline tongue lesions. Derived from the endoderm of the primitive foregut. Another theory is represented by the attachment of islands of endodermal cells from the floor of the primitive stomatodeum.	Ductal obstruction of mucous glands or embryological malformation
Polyhydramnios	±	±	N/A	±	±	N/A
US	Well-circumscribed, anechoic/homogeneously hypoechoic lesions; no Doppler signal	The presence of the thyroid gland is different from cystic swelling	N/A	Cystic and solid components commonly protruding out of the fetal mouth	To confirm the location of the mass. Smooth-surfaced and properly limited mass.	N/A
MRI	Presence of a ‘tail’ sign	N/A	Non-conclusive	Useful in patients with fistula and multilocular expansion	N/A	Prenatal diagnosis
Histo-pathology	Mucous cyst lined by salivary duct epithelium/remnants of ductal epithelium (columnar or cuboidal)	Non-keratinizing stratified columnar squamous, intermediate transitional, ± respiratory epithelium, ± thyroid follicles	Lymphoid aggregates—modified lymphatic channels; non-keratinized stratified epithelium	Keratinized, stratified squamous epithelium; heterogeneous masses with fat, cystic areas, and calcification.	Solid or cystic masses with gastric or intestinal epithelium with glands and muscle tissue in deep layers (foregut duplication cyst), ± respiratory epithelium (bronchogenic cyst)	Squamous epithelial may contain respiratory epithelium and mucous glands.
Therapeutic management	Needle aspiration, excision, marsupialization, cryo-surgery, resolve or rupture spontaneously. Observation with surgical intervention only if airway obstruction or feeding difficulties arise	Sistrunk’s operation	Follow-up/conservative surgical excision/marsupialization	Surgical removal	Surgery	Surgical decompression
Prognosis	Good	Malignancy in 1% of cases of thyroglossal duct cysts; no recurrence	Favorable prognosis; no recurrence	Potentially malignant; recurrence is extremely rare	Uncommon recurrence. Long-term follow-up is recommended	Recurrence is very low (in incomplete marsupialization)
References	[35,36,37]	[38,39,40]	[31,41]	[33,42]	[2,18,26,43,44]	[45,46,47]

## Data Availability

The data presented in the study are included in the article/Appendix A, further inquiries can be directed to the corresponding authors.

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
