# Peer review of "Mixed Heterotopic Gastrointestinal/Respiratory Oral Cysts in Newborns: From Prenatal Diagnosis to Histopathological and Therapeutic Management: A Case Report and Literature Review"

_diagnostics, 2024, doi:10.3390/diagnostics14030339_

Round 1

Reviewer 1 Report

Comments and Suggestions for Authors

The article is quite well written and novel in its concept.However a few points need to be rectified as follows:

1.The quality of written english needs to be improved

2.A few sentences are incoherent and not easily understandable and needs to be rephrased for a better understanding

3.The corrections required are highlighted in the attached manuscript.

Comments on the Quality of English Language

The article is quite well written and novel in its concept.However a few points need to be rectified as follows:

1.The quality of written english needs to be improved

2.A few sentences are incoherent and not easily understandable and needs to be rephrased for a better understanding

3.The corrections required are highlighted in the attached manuscript.

Author Response

The article is quite well written and novel in its concept. However a few points need to be rectified as follows:

1. The quality of written english needs to be improved

2. A few sentences are incoherent and not easily understandable and needs to be rephrased for a better understanding

3. The corrections required are highlighted in the attached manuscript.

Answer: Thank you for your valuable remarks. We corrected the article as you suggested, expanding the literature review and revising the English (please see the attached manuscript).

Kind regards

Reviewer 2 Report

Comments and Suggestions for Authors

This manuscript presents a significant and well-documented case, enriched by a detailed literature review. Your multidisciplinary approach and comprehensive case description are commendable. However, to meet the high standards of Diagnostics, a few areas require refinement.

- Please consider expanding the literature review to include the latest research, offering a more complete perspective on the subject.

- I recommend increasing the detail in the methodology section, particularly about diagnostic and therapeutic procedures, to enhance clarity and reproducibility.

- Consider elaborating on the uniqueness of this case in relation to existing literature, highlighting its novel insights or contrasts with previous studies.

- Please address potential limitations of your study in a dedicated section and suggest possible directions for future research.

Comments on the Quality of English Language

This manuscript presents valuable clinical information but is hindered by numerous language issues. These include inconsistent tense usage, grammatical errors, punctuation mistakes, and problems with clarity and technical language. A comprehensive language revision is recommended to bring the manuscript up to high publication standards.

Author Response

This manuscript presents a significant and well-documented case, enriched by a detailed literature review. Your multidisciplinary approach and comprehensive case description are commendable. However, to meet the high standards of Diagnostics, a few areas require refinement. Please consider expanding the literature review to include the latest research, offering a more complete perspective on the subject.

Answer: Thank you for your suggestion. We completed the literature review. (please see the attached manuscript). Table 1.

- I recommend increasing the detail in the methodology section, particularly about diagnostic and therapeutic procedures, to enhance clarity and reproducibility.

Answer: Thank you for your recommendation. We completed the manuscript (please see the attached manuscript).  Lines 152-164, 276-287

- Consider elaborating on the uniqueness of this case in relation to existing literature, highlighting its novel insights or contrasts with previous studies.

Answer: Thank you for your valuable remark. We mentioned the particular aspect of our case (please see the attached manuscript).  Lines 298-301

- Please address potential limitations of your study in a dedicated section and suggest possible directions for future research.

Answer: Thank you for your mention. We included the study's limitations and the directions for future research (please see the attached manuscript).  Lines 302-307

Kind regards

Round 2

Reviewer 2 Report

Comments and Suggestions for Authors

After a thorough review of your manuscript, it appears to be well-prepared and comprehensive, meeting the high standards necessary for publication. The depth of the case study, coupled with the literature review, presents a valuable contribution to the field. It seems the manuscript is now in a suitable form for publication.